# Improving Transferability of Representations via Augmentation-Aware Self-Supervision

**Hankook Lee**[1]   **Kibok Lee**[23*]   **Kimin Lee**[4]   **Honglak Lee**[25]   **Jinwoo Shin**[1]

[1]Korea Advanced Institute of Science and Technology (KAIST)
[2]University of Michigan
[3]Amazon Web Services
[4]University of California, Berkeley
[5]LG AI Research

## Abstract

Recent unsupervised representation learning methods have shown to be effective in a range of vision tasks by learning representations invariant to data augmentations such as random cropping and color jittering. However, such invariance could be harmful to downstream tasks if they rely on the characteristics of the data augmentations, e.g., location- or color-sensitive. This is not an issue just for unsupervised learning; we found that this occurs even in supervised learning because it also learns to predict the same label for all augmented samples of an instance. To avoid such failures and obtain more generalizable representations, we suggest to optimize an auxiliary self-supervised loss, coined *AugSelf*, that learns the difference of augmentation parameters (e.g., cropping positions, color adjustment intensities) between two randomly augmented samples. Our intuition is that AugSelf encourages to preserve augmentation-aware information in learned representations, which could be beneficial for their transferability. Furthermore, AugSelf can easily be incorporated into recent state-of-the-art representation learning methods with a negligible additional training cost. Extensive experiments demonstrate that our simple idea consistently improves the transferability of representations learned by supervised and unsupervised methods in various transfer learning scenarios. The code is available at https://github.com/hankook/AugSelf.

## 1   Introduction

Unsupervised representation learning has recently shown a remarkable success in various domains, e.g., computer vision [1, 2, 3], natural language [4, 5], code [6], reinforcement learning [7, 8, 9], and graphs [10]. The representations pretrained with a large number of unlabeled data have achieved outstanding performance in various downstream tasks, by either training task-specific layers on top of the model while freezing it or fine-tuning the entire model.

In the vision domain, the recent state-of-the-art methods [1, 2, 11, 12, 13] learn representations to be invariant to a pre-defined set of augmentations. The choice of the augmentations plays a crucial role in representation learning [2, 14, 15, 16]. A common choice is a combination of random cropping, horizontal flipping, color jittering, grayscaling, and Gaussian blurring. With this choice, learned representations are invariant to color and positional information in images; in other words, the representations lose such information.

On the contrary, there have also been attempts to learn representations by designing pretext tasks that keep such information in augmentations, e.g., predicting positional relations between two patches of

---

*Work done while at University of Michigan.

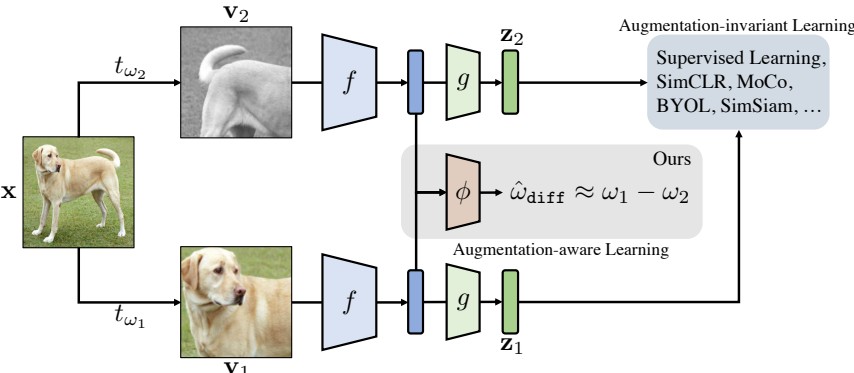

Figure 1: Illustration of the proposed method, AugSelf, that learns augmentation-aware information by predicting the difference between two augmentation parameters $\omega_1$ and $\omega_2$. Here, $\mathbf{x}$ is an original image, $\mathbf{v} = t_\omega(\mathbf{x})$ is an augmented sample by an augmentation $t_\omega$, $f$ is a feature extractor such as ResNet [20], and $g$ is a classifier for supervised learning or a projection MLP head for the recent unsupervised learning methods [1, 2, 12, 13].

an image [17], solving jigsaw puzzles [18], or predicting color information from a gray image [19]. These results show the importance of augmentation-specific information for representation learning, and inspire us to explore the following research questions: *when is learning invariance to a given set of augmentations harmful to representation learning?* and, *how to prevent the loss in the recent unsupervised learning methods?*

**Contribution.** We first found that learning representations with an augmentation-invariant objective might hurt its performance in downstream tasks that rely on information related to the augmentations. For example, learning invariance against strong color augmentations forces the representations to contain less color information (see Figure 2a). Hence, it degrades the performance of the representations in color-sensitive downstream tasks such as the Flowers classification task [21] (see Figure 2b).

To prevent this information loss and obtain more generalizable representations, we propose an auxiliary self-supervised loss, coined AugSelf, that learns the difference of augmentation parameters between the two augmented samples (or views) as shown in Figure 1. For example, in the case of random cropping, AugSelf learns to predict the difference of cropping positions of two randomly cropped views. We found that AugSelf encourages the self-supervised representation learning methods, such as SimCLR [2] and SimSiam [13], to preserve augmentation-aware information (see Figure 2a) that could be useful for downstream tasks. Furthermore, AugSelf can easily be incorporated into the recent unsupervised representation learning methods [1, 2, 12, 13] with a negligible additional training cost, which is for training an auxiliary prediction head $\phi$ in Figure 1.

Somewhat interestingly, we also found that optimizing the auxiliary loss, AugSelf, can even improve the transferability of representations learned under the standard supervised representation learning scenarios [22]. This is because supervised learning also forces invariance, i.e., assigns the same label, for all augmented samples (of the same instance), and AugSelf can help to keep augmentation-aware knowledge in the learned representations.

We demonstrate the effectiveness of AugSelf under extensive transfer learning experiments: AugSelf improves (a) two unsupervised representation learning methods, MoCo [1] and SimSiam [13], in 20 of 22 tested scenarios; (b) supervised pretraining in 9 of 11 (see Table 1). Furthermore, we found that AugSelf is also effective under few-shot learning setups (see Table 2).

Remark that learning augmentation-invariant representations has been a common practice for both supervised and unsupervised representation learning frameworks, while the importance of augmentation-awareness is less emphasized. We hope that our work could inspire researchers to rethink the under-explored aspect and provide a new angle in representation learning.

## 2 Preliminaries: Augmentation-invariant representation learning

In this section, we review the recent unsupervised representation learning methods [1, 2, 11, 12, 13] that learn representations by optimizing augmentation-invariant objectives. Formally, let $\mathbf{x}$ be an

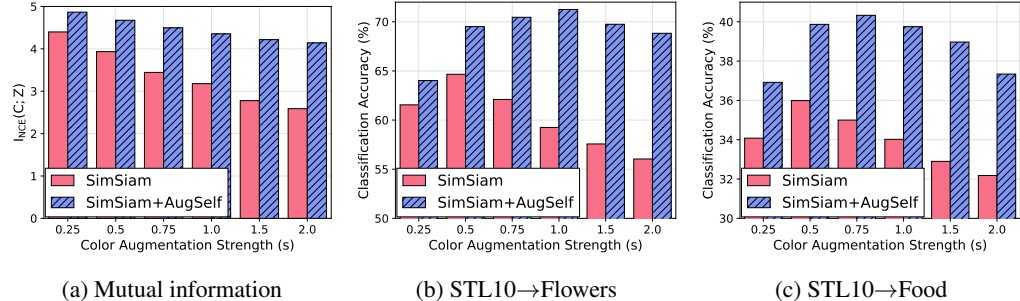

| (a) Mutual information | (b) STL10→Flowers | (c) STL10→Food |

Figure 2: (a) Changes of mutual information, i.e., $I_{\texttt{NCE}}(C; \mathbf{z})$, between color information $C(\mathbf{x})$ and the representation $\mathbf{z} = f(\mathbf{x})$ pretrained on STL10 [23] with varying the color jittering strength $s$. The pretrained representations are evaluated in color-sensitive benchmarks, (b) Flowers [21] and (c) Food [24], by the linear evaluation protocol [25].

image, $t_\omega$ be an augmentation function parameterized by an augmentation parameter $\omega$, $\mathbf{v} = t_\omega(\mathbf{x})$ be the augmented sample (or view) of $\mathbf{x}$ by $t_\omega$, and $f$ be a CNN feature extractor, such as ResNet [20]. Generally speaking, the methods encourage the representations $f(\mathbf{v}_1)$ and $f(\mathbf{v}_2)$ to be invariant to the two randomly augmented views $\mathbf{v}_1 = t_{\omega_1}(\mathbf{x})$ and $\mathbf{v}_2 = t_{\omega_2}(\mathbf{x})$, i.e., $f(\mathbf{v}_1) \approx f(\mathbf{v}_2)$ for $\omega_1, \omega_2 \sim \Omega$ where $\Omega$ is a pre-defined augmentation parameter distribution. We now describe the recent methods one by one briefly. For simplicity, we here omit the projection MLP head $g(\cdot)$ which is widely used in the methods (see Figure 1).

**Instance contrastive learning approaches** [1, 2, 26] minimize the distance between an anchor $f(t_{\omega_1}(\mathbf{x}))$ and its positive sample $f(t_{\omega_2}(\mathbf{x}))$, while maximizing the distance between the anchor $f(t_{\omega_1}(\mathbf{x}))$ and its negative sample $f(t_{\omega_3}(\mathbf{x}'))$. Since contrastive learning performance depends on the number of negative samples, a memory bank [26], a large batch [2], or a momentum network with a representation queue [1] has been utilized.

**Clustering approaches** [11, 27, 28] encourage two representations $f(t_{\omega_1}(\mathbf{x}))$ and $f(t_{\omega_2}(\mathbf{x}))$ to be assigned into the same cluster, in other words, the distance between them will be minimized.

**Negative-free methods** [12, 13] learn to predict the representation $f(\mathbf{v}_1)$ of a view $\mathbf{v}_1 = t_{\omega_1}(\mathbf{x})$ from another view $\mathbf{v}_2 = t_{\omega_2}(\mathbf{x})$. For example, SimSiam [13] minimizes $\|h(f(\mathbf{v}_2)) - \texttt{sg}(f(\mathbf{v}_1))\|_2^2$ where $h$ is an MLP and $\texttt{sg}$ is the stop-gradient operation. In these methods, if $h$ is optimal, then $h(f(\mathbf{v}_2)) = \mathbb{E}_{\omega_1 \sim \Omega}[f(\mathbf{v}_1)]$; thus, the expectation of the objective can be rewritten as $\text{Var}_{\omega \sim \Omega}(f(\mathbf{v}))$. Therefore, the methods can be considered as learning invariance with respect to the augmentations.

**Supervised learning approaches** [22] also learn augmentation-invariant representations. Since they often maximize $\exp(\mathbf{c}_y^\top f(t(\mathbf{x}))) / \sum_{y'} \exp(\mathbf{c}_{y'}^\top f(t(\mathbf{x})))$ where $\mathbf{c}_y$ is the prototype vector of the label $y$, $f(t(\mathbf{x}))$ is concentrated to $\mathbf{c}_y$, i.e., $\mathbf{c}_y \approx f(t_{\omega_1}(\mathbf{x})) \approx f(t_{\omega_2}(\mathbf{x}))$.

These approaches encourage representations $f(\mathbf{x})$ to contain shared (i.e., augmentation-invariant) information between $t_{\omega_1}(\mathbf{x})$ and $t_{\omega_2}(\mathbf{x})$ and discard other information [15]. For example, if $t_\omega$ changes color information, then to satisfy $f(t_{\omega_1}(\mathbf{x})) = f(t_{\omega_2}(\mathbf{x}))$ for any $\omega_1, \omega_2 \sim \Omega$, $f(\mathbf{x})$ will be learned to contain no (or less) color information. To verify this, we pretrain ResNet-18 [20] on STL10 [29] using SimSiam [13] with varying the strength $s$ of the color jittering augmentation. To measure the mutual information between representations and color information, we use the InfoNCE loss [30]. We here simply encode color information as RGB color histograms of an image. As shown in Figure 2a, using stronger color augmentations leads to color-relevant information loss. In classification on Flowers [21] and Food [24], which is color-sensitive, the learned representations containing less color information result in lower performance as shown in Figure 2b and 2c, respectively. This observation emphasizes the importance of learning augmentation-aware information in transfer learning scenarios.

## 3 Auxiliary augmentation-aware self-supervision

In this section, we introduce *auxiliary augmentation-aware self-supervision*, coined *AugSelf*, which encourages to preserve augmentation-aware information for generalizable representation learning. To

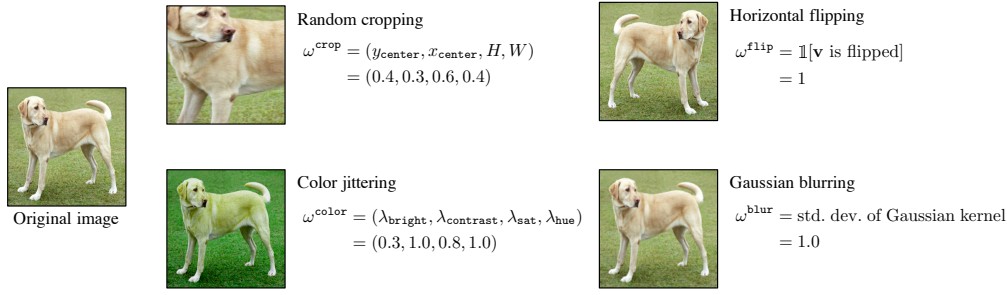

Figure 3: Examples of the commonly-used augmentations and their parameters $\omega^{\mathtt{aug}}$.

be specific, we add an auxiliary self-supervision loss, which learns to predict the difference between augmentation parameters of two randomly augmented views, into existing augmentation-invariant representation learning methods [1, 2, 11, 12, 13]. We first describe a general form of our auxiliary loss, and then specific forms for various augmentations. For conciseness, let $\theta$ be the collection of all parameters in the model.

Since an augmentation function $t_\omega$ is typically a composition of different types of augmentations, the augmentation parameter $\omega$ can be written as $\omega = (\omega^{\mathtt{aug}})_{\mathtt{aug}\in\mathcal{A}}$ where $\mathcal{A}$ is the set of augmentations used in pretraining (e.g., $\mathcal{A} = \{\mathtt{crop}, \mathtt{flip}\}$), and $\omega^{\mathtt{aug}}$ is an augmentation-specific parameter (e.g., $\omega^{\mathtt{crop}}$ decides how to crop an image). Then, given two randomly augmented views $\mathbf{v}_1 = t_{\omega_1}(\mathbf{x})$ and $\mathbf{v}_2 = t_{\omega_2}(\mathbf{x})$, the AugSelf objective is as follows:

$$\mathcal{L}_{\mathtt{AugSelf}}(\mathbf{x},\omega_1,\omega_2;\theta) = \sum\nolimits_{\mathtt{aug}\in\mathcal{A}_{\mathtt{AugSelf}}} \mathcal{L}_{\mathtt{aug}}\big(\phi_\theta^{\mathtt{aug}}(f_\theta(\mathbf{v}_1), f_\theta(\mathbf{v}_2)), \omega_{\mathtt{diff}}^{\mathtt{aug}}\big),$$

where $\mathcal{A}_{\mathtt{AugSelf}} \subseteq \mathcal{A}$ is the set of augmentations for augmentation-aware learning, $\omega_{\mathtt{diff}}^{\mathtt{aug}}$ is the difference between two augmentation-specific parameters $\omega_1^{\mathtt{aug}}$ and $\omega_2^{\mathtt{aug}}$, $\mathcal{L}_{\mathtt{aug}}$ is an augmentation-specific loss, and $\phi_\theta^{\mathtt{aug}}$ is a 3-layer MLP for $\omega_{\mathtt{diff}}^{\mathtt{aug}}$ prediction. This design allows us to incorporate AugSelf into the recent state-of-the-art unsupervised learning methods [1, 2, 11, 12, 13] with a negligible additional training cost. For example, the objective of SimSiam [13] with AugSelf can be written as $\mathcal{L}_{\mathtt{total}}(\mathbf{x},\omega_1,\omega_2;\theta) = \mathcal{L}_{\mathtt{SimSiam}}(\mathbf{x},\omega_1,\omega_2;\theta) + \lambda \cdot \mathcal{L}_{\mathtt{AugSelf}}(\mathbf{x},\omega_1,\omega_2;\theta)$, where $\lambda$ is a hyperparameter for balancing losses. Remark that the total objective $\mathcal{L}_{\mathtt{total}}$ encourages the shared representation $f(\mathbf{x})$ to learn both augmentation-invariant and augmentation-aware features. Hence, the learned representation $f(\mathbf{x})$ also can be useful in various downstream (e.g., augmentation-sensitive) tasks.[2]

In this paper, we mainly focus on the commonly-used augmentations in the recent unsupervised representation learning methods [1, 2, 11, 12, 13]: random cropping, random horizontal flipping, color jittering, and Gaussian blurring; however, we remark that different types of augmentations can be incorporated into AugSelf (see Section 4.2). In the following, we elaborate on the details of $\omega^{\mathtt{aug}}$ and $\mathcal{L}_{\mathtt{aug}}$ for each augmentation. The examples of $\omega^{\mathtt{aug}}$ are illustrated in Figure 3.

**Random cropping.** The random cropping is the most popular augmentation in vision tasks. A cropping parameter $\omega^{\mathtt{crop}}$ contains the center position and cropping size. We normalize the values by the height and width of the original image $\mathbf{x}$, i.e., $\omega^{\mathtt{crop}} \in [0,1]^4$. Then, we use $\ell_2$ loss for $\mathcal{L}_{\mathtt{crop}}$ and set $\omega_{\mathtt{diff}}^{\mathtt{crop}} = \omega_1^{\mathtt{crop}} - \omega_2^{\mathtt{crop}}$.

**Random horizontal flipping.** A flipping parameter $\omega^{\mathtt{flip}} \in \{0,1\}$ indicates the image is horizontally flipped or not. Since it is discrete, we use the binary cross-entropy loss for $\mathcal{L}_{\mathtt{flip}}$ and set $\omega_{\mathtt{diff}}^{\mathtt{flip}} = \mathbb{1}[\omega_1^{\mathtt{flip}} = \omega_2^{\mathtt{flip}}]$.

**Color jittering.** The color jittering augmentation adjusts brightness, contrast, saturation, and hue of an input image in a random order. For each adjustment, its intensity is uniformly sampled from a pre-defined interval. We normalize all intensities into $[0,1]$, i.e., $\omega^{\mathtt{color}} \in [0,1]^4$. Similarly to cropping, we use $\ell_2$ loss for $\mathcal{L}_{\mathtt{color}}$ and set $\omega_{\mathtt{diff}}^{\mathtt{color}} = \omega_1^{\mathtt{color}} - \omega_2^{\mathtt{color}}$.

---

[2]We observe that our augmentation-aware objective $\mathcal{L}_{\mathtt{AugSelf}}$ does not interfere with learning the augmentation-invariant objective, e.g., $\mathcal{L}_{\mathtt{SimSiam}}$. This allows $f(\mathbf{x})$ to learn augmentation-aware information with a negligible loss of augmentation-invariant information. A detailed discussion is provided in the supplementary material.

Table 1: Linear evaluation accuracy (%) of ResNet-50 [20] and ResNet-18 pretrained on ImageNet100 [31, 32] and STL10 [23], respectively. **Bold entries** are the best of each baseline.

| Method | CIFAR10 | CIFAR100 | Food | MIT67 | Pets | Flowers | Caltech101 | Cars | Aircraft | DTD | SUN397 |
|---|---|---|---|---|---|---|---|---|---|---|---|
| *ImageNet100-pretrained ResNet-50* | | | | | | | | | | | |
| SimSiam | 86.89 | 66.33 | 61.48 | 65.75 | 74.69 | 88.06 | 84.13 | **48.20** | 48.63 | 65.11 | 50.60 |
| + AugSelf (ours) | **88.80** | **70.27** | **65.63** | **67.76** | **76.34** | **90.70** | **85.30** | 47.52 | **49.76** | **67.29** | **52.28** |
| MoCo v2 | 84.60 | 61.60 | 59.37 | 61.64 | 70.08 | 82.43 | 77.25 | 33.86 | **41.21** | 64.47 | 46.50 |
| + AugSelf (ours) | **85.26** | **63.90** | **60.78** | **63.36** | **73.46** | **85.70** | **78.93** | **37.35** | 39.47 | **66.22** | **48.52** |
| Supervised | **86.16** | 62.70 | 53.89 | 52.91 | 73.50 | 76.09 | **77.53** | 30.61 | 36.78 | 61.91 | 40.59 |
| + AugSelf (ours) | 86.06 | **63.77** | **55.84** | **54.63** | **74.81** | **78.22** | 77.47 | **31.26** | **38.02** | **62.07** | **41.49** |
| *STL10-pretrained ResNet-18* | | | | | | | | | | | |
| SimSiam | 82.35 | 54.90 | 33.99 | 39.15 | 44.90 | 59.19 | 66.33 | 16.85 | 26.06 | 42.57 | 29.05 |
| + AugSelf (ours) | **82.76** | **58.65** | **41.58** | **45.67** | **48.42** | **72.18** | **72.75** | **21.17** | **33.17** | **47.02** | **34.14** |
| MoCo v2 | 81.18 | 53.75 | 33.69 | 39.01 | 42.34 | 61.01 | 64.15 | 16.09 | 26.63 | 41.20 | 28.50 |
| + AugSelf (ours) | **82.45** | **57.17** | **36.91** | **41.67** | **43.80** | **66.96** | **66.02** | **17.53** | **28.02** | **45.21** | **30.93** |

**Gaussian blurring.** This blurring operation is widely used in unsupervised representation learning. The Gaussian filter is constructed by a single parameter, standard deviation $\sigma = \omega^{\texttt{blur}}$. We also normalize the parameter into $[0, 1]$. Then, we use $\ell_2$ loss for $\mathcal{L}_{\texttt{blur}}$ and set $\omega^{\texttt{blur}}_{\texttt{diff}} = \omega^{\texttt{blur}}_1 - \omega^{\texttt{blur}}_2$.

## 4 Experiments

**Setup.** We pretrain the standard ResNet-18 [20] and ResNet-50 on STL10 [23] and ImageNet100[3] [31, 32], respectively. We use two recent unsupervised representation learning methods as baselines for pretraining: a contrastive method, MoCo v2 [1, 14], and a non-contrastive method, SimSiam [13]. For STL10 and ImageNet100, we pretrain networks for 200 and 500 epochs with a batch size of 256, respectively. For supervised pretraining, we pretrain ResNet-50 for 100 epochs with a batch size of 128 on ImageNet100.[4] For augmentations, we use random cropping, flipping, color jittering, grayscaling, and Gaussian blurring following Chen and He [13]. In this section, our AugSelf predicts random cropping and color jittering parameters, i.e., $\mathcal{A}_{\texttt{AugSelf}} = \{\texttt{crop}, \texttt{color}\}$, unless otherwise stated. We set $\lambda = 1.0$ for STL10 and $\lambda = 0.5$ for ImageNet100. The other details and the sensitivity analysis to the hyperparameter $\lambda$ are provided in the supplementary material. For ablation study (Section 4.2), we only use STL10-pretrained models.

### 4.1 Main results

**Linear evaluation in various downstream tasks.** We evaluate the pretrained networks in downstream classification tasks on 11 datasets: CIFAR10/100 [29], Food [24], MIT67 [36], Pets [37], Flowers [21], Caltech101 [38], Cars [39], Aircraft [40], DTD [41], and SUN397 [42]. They contain roughly 1k∼70k training images. We follow the linear evaluation protocol [25]. The detailed information of datasets and experimental settings is described in the supplementary material. Table 1 shows the transfer learning results in the various downstream tasks. Our AugSelf consistently improves (a) the recent unsupervised representation learning methods, SimSiam [13] and MoCo [13], in 10 out of 11 downstream tasks; and (b) supervised pretraining in 9 out of 11 downstream tasks. These consistent improvements imply that our method encourages to learn more generalizable representations.

**Few-shot classification.** We also evaluate the pretrained networks on various few-shot learning benchmarks: FC100 [33], Caltech-UCSD Birds (CUB200) [43], and Plant Disease [35]. Note that CUB200 and Plant Disease benchmarks require low-level features such as color information of birds and leaves, respectively, to detect their fine-grained labels. They are widely used in cross-domain few-shot settings [44, 45]. For few-shot learning, we perform logistic regression using the frozen representations $f(\mathbf{x})$ without fine-tuning. Table 2 shows the few-shot learning performance of 5-way 1-shot and 5-way 5-shot tasks. As shown in the table, our AugSelf improves the performance of SimSiam [13] and MoCo [1] in all cases with a large margin. For example, for plant disease detection

---

[3]ImageNet100 is a 100-category subset of ImageNet [31]. We use the same split following Tian et al. [32].

[4]We do not experiment supervised pretraining on STL10, as it has only 5k labeled training samples, which is not enough for pretraining a good representation.

Table 2: Few-shot classification accuracy (%) with 95% confidence intervals averaged over 2000 episodes on FC100 [33], CUB200 [34], and Plant Disease [35]. $(N, K)$ denotes $N$-way $K$-shot tasks. **Bold entries** are the best of each group.

| Method | FC100 | | CUB200 | | Plant Disease | |
|---|---|---|---|---|---|---|
| | (5, 1) | (5, 5) | (5, 1) | (5, 5) | (5, 1) | (5, 5) |
| *ImageNet100-pretrained ResNet-50* | | | | | | |
| SimSiam | $36.19_{\pm0.36}$ | $50.36_{\pm0.38}$ | $45.56_{\pm0.47}$ | $62.48_{\pm0.48}$ | $75.72_{\pm0.46}$ | $89.94_{\pm0.31}$ |
| + AugSelf (ours) | $\mathbf{39.37_{\pm0.40}}$ | $\mathbf{55.27_{\pm0.38}}$ | $\mathbf{48.08_{\pm0.47}}$ | $\mathbf{66.27_{\pm0.46}}$ | $\mathbf{77.93_{\pm0.46}}$ | $\mathbf{91.52_{\pm0.29}}$ |
| MoCo v2 | $31.67_{\pm0.33}$ | $43.88_{\pm0.38}$ | $41.67_{\pm0.47}$ | $56.92_{\pm0.47}$ | $65.73_{\pm0.49}$ | $84.98_{\pm0.36}$ |
| + AugSelf (ours) | $\mathbf{35.02_{\pm0.36}}$ | $\mathbf{48.77_{\pm0.39}}$ | $\mathbf{44.17_{\pm0.48}}$ | $\mathbf{57.35_{\pm0.48}}$ | $\mathbf{71.80_{\pm0.47}}$ | $\mathbf{87.81_{\pm0.33}}$ |
| Supervised | $33.15_{\pm0.33}$ | $46.59_{\pm0.37}$ | $46.57_{\pm0.48}$ | $63.69_{\pm0.46}$ | $68.95_{\pm0.47}$ | $88.77_{\pm0.30}$ |
| + AugSelf (ours) | $\mathbf{34.70_{\pm0.35}}$ | $\mathbf{48.89_{\pm0.38}}$ | $\mathbf{47.58_{\pm0.48}}$ | $\mathbf{65.31_{\pm0.45}}$ | $\mathbf{70.82_{\pm0.46}}$ | $\mathbf{89.77_{\pm0.29}}$ |
| *STL10-pretrained ResNet-18* | | | | | | |
| SimSiam | $36.72_{\pm0.35}$ | $51.49_{\pm0.36}$ | $37.97_{\pm0.43}$ | $50.61_{\pm0.45}$ | $58.13_{\pm0.50}$ | $75.98_{\pm0.40}$ |
| + AugSelf (ours) | $\mathbf{40.68_{\pm0.39}}$ | $\mathbf{56.26_{\pm0.38}}$ | $\mathbf{41.60_{\pm0.42}}$ | $\mathbf{56.33_{\pm0.44}}$ | $\mathbf{62.85_{\pm0.49}}$ | $\mathbf{81.14_{\pm0.37}}$ |
| MoCo v2 | $35.69_{\pm0.34}$ | $49.26_{\pm0.36}$ | $37.62_{\pm0.42}$ | $50.71_{\pm0.44}$ | $57.87_{\pm0.48}$ | $75.98_{\pm0.40}$ |
| + AugSelf (ours) | $\mathbf{39.66_{\pm0.39}}$ | $\mathbf{55.58_{\pm0.39}}$ | $\mathbf{38.33_{\pm0.41}}$ | $\mathbf{51.93_{\pm0.44}}$ | $\mathbf{60.78_{\pm0.50}}$ | $\mathbf{78.76_{\pm0.38}}$ |

Table 3: Linear evaluation accuracy (%) under the same setup following Xiao et al. [16]. The augmentations in the brackets of LooC [16] indicate which augmentation-aware information is learned. $N$ is the number of required augmented samples for each instance, that reflects the effective training batch size. $*$ indicates that the numbers are reported in [16]. The numbers in the brackets show the accuracy gains compared to each baseline.

| Method | $N$ | ImageNet100 | CUB200 | Flowers (5-shot) | Flowers (10-shot) |
|---|---|---|---|---|---|
| MoCo* [1] | 2 | 81.0 | 36.7 | $67.9_{\pm0.5}$ | $77.3_{\pm0.1}$ |
| LooC* [16] (color) | 3 | 81.1 (+0.1) | 40.1 (+3.4) | $68.2_{\pm0.6}$ (+0.3) | $77.6_{\pm0.1}$ (+0.3) |
| LooC* [16] (rotation) | 3 | 80.2 (-0.8) | 38.8 (+2.1) | $70.1_{\pm0.4}$ (+2.2) | $79.3_{\pm0.1}$ (+2.0) |
| LooC* [16] (color, rotation) | 4 | 79.2 (-1.8) | 39.6 (+2.9) | $70.9_{\pm0.3}$ (+3.0) | $80.8_{\pm0.2}$ (+3.5) |
| MoCo [1] | 2 | 81.0 | 32.2 | $78.5_{\pm0.3}$ | $81.2_{\pm0.3}$ |
| MoCo [1] + AugSelf (ours) | 2 | 82.4 (+1.4) | 37.0 (+4.8) | $81.7_{\pm0.2}$ (+3.2) | $84.5_{\pm0.2}$ (+3.3) |
| SimSiam [13] | 2 | 81.6 | 38.4 | $83.6_{\pm0.3}$ | $85.9_{\pm0.2}$ |
| SimSiam [13] + AugSelf (ours) | 2 | $\mathbf{82.6}$ (+1.0) | $\mathbf{45.3}$ (+6.9) | $\mathbf{86.4_{\pm0.2}}$ (+2.8) | $\mathbf{88.3_{\pm0.1}}$ (+2.4) |

[35], we obtain up to 6.07% accuracy gain in 5-way 1-shot tasks. These results show that our method is also effective in such transfer learning scenarios.

**Comparison with LooC.** Recently, Xiao et al. [16] propose LooC that learns augmentation-aware representations via multiple augmentation-specific contrastive learning objectives. Table 3 shows head-to-head comparisons under the same evaluation setup following Xiao et al. [16].[5] As shown in the table, our AugSelf has two advantages over LooC: (a) AugSelf requires the same number of augmented samples compared to the baseline unsupervised representation learning methods while LooC requires more, such that AugSelf does not increase the computational cost; (b) AugSelf can be incorporated with non-contrastive methods e.g., SimSiam [13], and SimSiam with AugSelf outperforms LooC in all cases.

**Object localization.** We also evaluate representations in an object localization task (i.e., bounding box prediction) that requires positional information. We experiment on CUB200 [46] and solve linear regression using representations pretrained by SimSiam [13] without or with our method. Table 4 reports $\ell_2$ errors of bounding box predictions and Figure 4 shows the examples of the predictions. These results demonstrate that AugSelf is capable of learning positional information.

**Retrieval.** Figure 5 shows the retrieval results using pretrained models. For this experiment, we use the Flowers [21] and Cars [39] datasets and find top-4 nearest neighbors based on the cosine

---

[5]Since LooC's code is currently not publicly available, we reproduced the MoCo baseline as reported in the sixth row in Table 3: we obtained the same ImageNet100 result, but different ones for CUB200 and Flowers.

| Method | Error |
|---|---|
| SimSiam | 0.00462 |
| + AugSelf (ours) | **0.00335** |
| MoCo | 0.00487 |
| + AugSelf (ours) | **0.00429** |
| Supervised | 0.00520 |
| + AugSelf (ours) | **0.00473** |

Table 4: $\ell_2$ errors of bounding box predictions on CUB200.

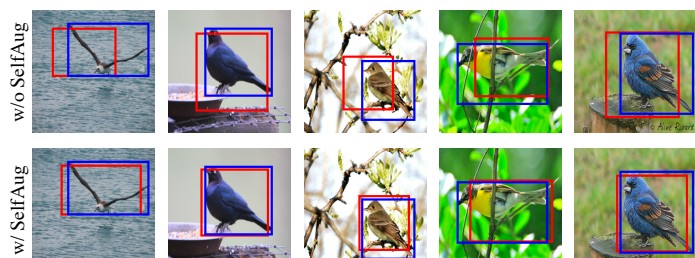

Figure 4: Examples of bounding box predictions on CUB200. Blue and red boxes are ground-truth and model prediction, respectively.

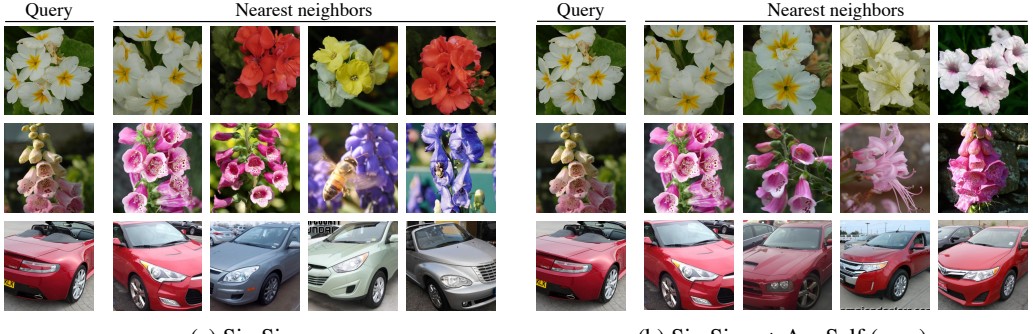

(a) SimSiam       (b) SimSiam + AugSelf (ours)

Figure 5: Top-4 nearest neighbors based on the cosine similarity using representations $f(\mathbf{x})$ learned by (a) SimSiam [13] or (b) SimSiam with AugSelf (ours).

similarity between representations $f(\mathbf{x})$ where $f$ is the pretrained ResNet-50 on ImageNet100. As shown in the figure, the representations learned by AugSelf are more color-sensitive.

## 4.2 Ablation study

**Effect of augmentation prediction tasks.** We first evaluate the proposed augmentation prediction tasks one by one without incorporating invariance-learning methods. More specifically, we pretrain $f_\theta$ using only $\mathcal{L}_{\texttt{aug}}$ for each $\texttt{aug} \in \{\texttt{crop}, \texttt{flip}, \texttt{color}, \texttt{blur}\}$. Remark that training objectives are different but we use the same set of augmentations. Table 5 shows the transfer learning results in various downstream tasks. We observe that solving horizontal flipping and Gaussian blurring prediction tasks results in worse or similar performance to a random initialized network in various downstream tasks, i.e., the augmentations do not contain task-relevant information. However, solving random cropping and color jittering prediction tasks significantly outperforms the random initialization in all downstream tasks. Furthermore, surprisingly, the color jittering prediction task achieves competitive performance in the Flowers [21] dataset compared to a recent state-of-the-art method, SimSiam [13]. These results show that augmentation-aware information are task-relevant and learning such information could be important in downstream tasks.

Based on the above observations, we incorporate random cropping and color jittering prediction tasks into SimSiam [13] when pretraining. More specifically, we optimize $\mathcal{L}_{\texttt{SimSiam}} + \lambda_{\texttt{crop}}\mathcal{L}_{\texttt{crop}} + \lambda_{\texttt{color}}\mathcal{L}_{\texttt{color}}$ where $\lambda_{\texttt{crop}}, \lambda_{\texttt{color}} \in \{0, 1\}$. The transfer learning results are reported in Table 6. As shown in the table, each self-supervision task improves SimSiam consistently (and often significantly) in various downstream tasks. For example, the color jittering prediction task improves SimSiam by 6.33% and 11.89% in Food [24] and Flowers [21] benchmarks, respectively. When incorporating both tasks simultaneously, we achieve further improvements in almost all the downstream tasks. Furthermore, as shown in Figure 2, our AugSelf preserves augmentation-aware information as much as possible; hence our gain is consistent regardless of the strength of color jittering augmentation.

**Different augmentations.** We confirm that our method can allow to use other strong augmentations: rotation, which rotates an image by $0°, 90°, 180°, 270°$ degrees randomly; and solarization, which inverts each pixel value when the value is larger than a randomly sampled threshold. Based on the default augmentation setting, i.e., $\mathcal{A}_{\texttt{AugSelf}} = \{\texttt{crop}, \texttt{color}\}$, we additionally apply each

Table 5: Linear evaluation accuracy (%) of ResNet-18 [20] pretrained by each augmentation prediction task without other methods such as SimSiam [13]. We report SimSiam [13] results as reference. **Bold entries** are larger than the random initialization.

| Pretraining objective | STL10 | CIFAR10 | CIFAR100 | Food | MIT67 | Pets | Flowers |
|---|---|---|---|---|---|---|---|
| Random Init | 42.72 | 47.45 | 23.73 | 11.54 | 12.29 | 12.94 | 26.06 |
| $\mathcal{L}_{\texttt{crop}}$ | **68.28** | **70.78** | **43.44** | **22.26** | **26.17** | **27.68** | **38.21** |
| $\mathcal{L}_{\texttt{flip}}$ | **46.45** | **53.80** | **24.89** | 9.69 | 11.99 | 10.71 | 13.04 |
| $\mathcal{L}_{\texttt{color}}$ | **61.14** | **63.39** | **40.38** | **28.02** | **25.35** | **24.49** | **54.42** |
| $\mathcal{L}_{\texttt{blur}}$ | **48.26** | 46.60 | 20.44 | 8.73 | 11.87 | **13.07** | 17.20 |
| SimSiam [13] | **85.19** | **82.35** | **54.90** | **33.99** | **39.15** | **44.90** | **59.19** |

Table 6: Linear evaluation accuracy (%) of ResNet-18 [20] pretrained by SimSiam [13] with various combinations of our augmentation prediction tasks. **Bold entries** are the best of each task.

| $\mathcal{A}_{\texttt{AugSelf}}$ | STL10 | CIFAR10 | CIFAR100 | Food | MIT67 | Pets | Flowers |
|---|---|---|---|---|---|---|---|
| $\emptyset$ | 85.19 | 82.35 | 54.90 | 33.99 | 39.15 | 44.90 | 59.19 |
| $\{\texttt{crop}\}$ | **85.98** | 82.82 | 55.78 | 35.68 | 43.21 | 47.10 | 62.05 |
| $\{\texttt{color}\}$ | 85.55 | **82.90** | 58.11 | 40.32 | 43.56 | 47.85 | 71.08 |
| $\{\texttt{crop,color}\}$ | 85.70 | 82.76 | **58.65** | **41.58** | **45.67** | **48.42** | **72.18** |

augmentation with a probability of 0.5. We also evaluate the effectiveness of augmentation prediction tasks for rotation and solarization. Note that we formulate the rotation prediction as a 4-way classification task (i.e., $\omega_{\texttt{diff}}^{\texttt{rot}} \in \{0, 1, 2, 3\}$) and the solarization prediction as a regression task (i.e., $\omega_{\texttt{diff}}^{\texttt{sol}} \in [-1, 1]$). As shown in Table 7, we obtain consistent gains across various downstream tasks even if stronger augmentations are applied. Furthermore, in the case of rotation, we observe that our augmentation prediction task tries to prevent the performance degradation from learning invariance to rotations. For example, in CIFAR100 [29], the baseline loses 4.66% accuracy (54.90%→50.24%) when using rotations, but ours does only 0.37% (58.65%→58.28%). These results show that our AugSelf is less sensitive to the choice of augmentations. We believe that this robustness would be useful in future research on representation learning with strong augmentations.

**Solving geometric and color-related pretext tasks.** To validate that our AugSelf is capable of learning augmentation-aware information, we try to solve two pretext tasks requiring the information: 4-way rotation ($0°, 90°, 180°, 270°$) and 6-way color channel permutation (RGB, RBG, ..., BGR) classification tasks. We note that the baseline (SimSiam) and our method (SimSiam+AugSelf) do not observe rotated or

Table 8: Linear evaluation accuracy in augmentation-aware pretext tasks.

| Method | Rotation | Color perm |
|---|---|---|
| SimSiam | 59.11 | 24.66 |
| + AugSelf | 64.61 | 60.49 |

color-permuted samples in the pretraining phase. We train a linear classifier on top of pretrained representation without finetuning for each task. As reported in Table 8, our AugSelf solves the pretext tasks well even without their prior knowledge in pretraining; these results validate that our method learns augmentation-aware information.

**Compatibility with other methods.** While we mainly focus on SimSiam [13] and MoCo [1] in the previous section, our AugSelf can be incorporated into other unsupervised learning methods, SimCLR [2], BYOL [12], and SwAV [11]. Table 9 shows the consistent and significant gains by AugSelf across all methods and downstream tasks.

## 5 Related work

**Self-supervised pretext tasks.** For visual representation learning without labels, various pretext tasks have been proposed in literature [17, 18, 19, 47, 48, 49, 50, 51] by constructing self-supervision from an image. For example, Doersch et al. [17], Noroozi and Favaro [18] split the original image x into $3 \times 3$ patches and then learn visual representations by predicting relations between the patch locations. Instead, Zhang et al. [19], Larsson et al. [48] construct color prediction tasks by converting colorful images to gray ones. Zhang et al. [51] propose a similar task requiring to predict one subset of channels (e.g., depth) from another (e.g., RGB values). Meanwhile, Gidaris et al. [47], Qi et al.

Table 7: Transfer learning accuracy (%) of ResNet-18 [20] pretrained by SimSiam [13] with or without our AugSelf using strong augmentations. C, J, R and S denote cropping, color jittering, rotation and solarization prediction tasks, respectively. **Bold entries** are the best of each augmentation.

| Strong Aug. | $\mathcal{A}_{\texttt{AugSelf}}$ | STL10 | CIFAR10 | CIFAR100 | Food | MIT67 | Pets | Flowers |
|---|---|---|---|---|---|---|---|---|
| None | $\emptyset$ | 85.19 | 82.35 | 54.90 | 33.99 | 39.15 | 44.90 | 59.19 |
| | {C, J} | **85.70** | **82.76** | **58.65** | **41.58** | **45.67** | **48.42** | **72.18** |
| Rotation | $\emptyset$ | 80.11 | 77.78 | 50.24 | 36.40 | 36.39 | 41.43 | 61.77 |
| | {C, J} | 81.85 | 79.93 | 57.27 | 43.04 | 41.32 | **47.30** | 72.52 |
| | {C, J, R} | **82.67** | **80.71** | **58.28** | **43.28** | **44.48** | 46.65 | **72.94** |
| Solarization | $\emptyset$ | **86.32** | 81.08 | 52.50 | 32.59 | 41.29 | 44.76 | 58.79 |
| | {C, J} | 86.03 | **82.64** | 57.94 | **40.29** | **46.67** | 48.81 | **71.43** |
| | {C, J, S} | 85.91 | 82.63 | **58.18** | 40.17 | 45.57 | **49.02** | **71.43** |

Table 9: Transfer learning accuracy (%) of various unsupervised learning frameworks with and without our AugSelf framework. **Bold entries** indicates the best for each baseline method.

| Method | AugSelf (ours) | STL10 | CIFAR10 | CIFAR100 | Food | MIT67 | Pets | Flowers |
|---|---|---|---|---|---|---|---|---|
| SimCLR [2] | | 84.87 | 78.93 | 48.94 | 31.97 | 36.82 | 43.18 | 56.20 |
| | ✓ | **84.99** | **80.92** | **53.64** | **36.21** | **40.62** | **46.51** | **64.31** |
| BYOL [12] | | 86.73 | 82.66 | 55.94 | 37.30 | 42.78 | 50.21 | 66.89 |
| | ✓ | **86.79** | **83.60** | **59.66** | **42.89** | **46.17** | **52.45** | **74.07** |
| SwAV [11] | | 82.21 | 81.60 | 52.00 | 29.78 | 36.69 | 37.68 | 53.01 |
| | ✓ | **82.57** | **82.00** | **55.10** | **33.16** | **39.13** | **40.74** | **61.69** |

[49], Zhang et al. [50] show that solving affine transformation (e.g., rotation) prediction tasks can learn high-level representations. These approaches often require specific preprocessing procedures (e.g., $3 \times 3$ patches [17, 18], or specific affine transformations [49, 50]). In contrast, our AugSelf can be working with common augmentations such as random cropping and color jittering. This advantage allows us to incorporate AugSelf to the recent state-of-the-art frameworks like SimCLR [2] while not increasing the computational cost. Furthermore, we emphasize that our contribution is not only to construct AugSelf, but also finding on the importance of learning augmentation-aware representations together with the existing augmentation-invariant approaches.

**Augmentations for unsupervised representation learning.** Chen et al. [2], Tian et al. [15] found that the choice of augmentations plays a critical role in contrastive learning. Based on this finding, many unsupervised learning methods [1, 2, 11, 12, 13] have used similar augmentations (e.g., cropping and color jittering) and then achieved outstanding performance in ImageNet [31]. Tian et al. [15] discuss a similar observation to ours that the optimal choice of augmentations is task-dependent, but they focus on finding the optimal choice for a specific downstream task. However, in the pretraining phase, prior knowledge of downstream tasks could be not available. In this case, we need to preserve augmentation-aware information in representations as much as possible for unknown downstream tasks. Recently, Xiao et al. [16] propose a contrastive method for learning augmentation-aware representations. This method requires additional augmented samples for each augmentation; hence, the training cost increases with respect to the number of augmentations. Furthermore, the method is specialized to contrastive learning only, and is not attractive to be used for non-contrastive methods like BYOL [12]. In contrast, our AugSelf does not suffer from these issues as shown in Table 3 and 9.

## 6 Discussion and conclusion

To improve the transferability of representations, we propose AugSelf, an auxiliary augmentation-aware self-supervision method that encourages representations to contain augmentation-aware information that could be useful in downstream tasks. Our idea is to learn to predict the difference between augmentation parameters of two randomly augmented views. Through extensive experiments, we demonstrate the effectiveness of AugSelf in various transfer learning scenarios. We believe that our work would guide many research directions for unsupervised representation learning and transfer learning.

**Limitations.** Even though our method provides large gains when we apply it to popular data augmentations like random cropping, it might not be applicable to some specific data augmentation, where it is non-trivial to parameterize augmentation such as GAN-based one [52]. Hence, an interesting future direction would be to develop an augmentation-agnostic way that does not require to explicitly design augmentation-specific self-supervision. Even for this direction, we think that our idea of *learning the relation between two augmented samples* could be used, e.g., constructing contrastive pairs between the relations.

**Negative societal impacts.** Self-supervised training typically requires a huge training cost (e.g., training MoCo v2 [14] for 1000 epochs requires 11 days on 8 V100 GPUs), a large network capacity (e.g., GPT3 [5] requires 175 billion parameters), therefore it raises environmental issues such as carbon generation [53]. Hence, efficient training methods [54] or distilling knowledge to a smaller network [55] would be required to ameliorate such environmental problems.

## Acknowledgments and disclosure of funding

This work was mainly supported by Samsung Electronics Co., Ltd (IO201211-08107-01) and partly supported by Institute of Information & Communications Technology Planning & Evaluation (IITP) grant funded by the Korea government (MSIT) (No.2019-0-00075, Artificial Intelligence Graduate School Program (KAIST)).

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
