# Supplementary Material:
# Improving Transferability of Representations via Augmentation-Aware Self-Supervision

## A    Trade-off between augmentation invariance and awareness

We first emphasize that we want $f(\mathbf{x})$ to learn both augmentation-invariant and augmentation-aware information (or features) in the input $\mathbf{x}$. To this end, we train $g$ and $\phi$ to extract each information from $f(\mathbf{x})$, respectively; in other words, we want the functions $g(f(t_1(\mathbf{x})))$ and $\phi(f(t_1(\mathbf{x})), f(t_2(\mathbf{x})))$ to be invariant and variant with respect to the augmentation $t_1$ (and $t_2$), respectively. Here, if the shared network $f$ has a limited capacity (e.g., few parameters or dimension), the two training objectives (for $g$ and $\phi$) may interfere with each other, i.e., $f(\mathbf{x})$ might become less invariant (or contain less augmentation-invariant information). However, our choice $f$ of deep neural networks (DNNs) in our experiments does not suffer from the issue (i.e., DNNs are highly expressive), so our goal is achievable with a negligible loss of augmentation-invariant information. To support this, we compute the cosine similarity between representations from augmented and original samples, i.e., $\mathtt{CS} = \mathbb{E}_{\mathbf{x}\sim\mathcal{D},t\sim\mathcal{T}}[\mathtt{sim}(g \circ f(t(\mathbf{x})), g \circ f(\mathbf{x}))]$. Note that this metric becomes higher as the representation $g(f(\mathbf{x}))$ is more invariant to the augmentation $t \sim \mathcal{T}$. We here use STL10-pretrained models. Table 1 shows that AugSelf does not significantly change the cosine similarity $\mathtt{CS}$; in other words, AugSelf is not harmful to the augmentation-invariant objective.

Table 1: The invariance metric, $\mathtt{CS} = \mathbb{E}_{\mathbf{x}\sim\mathcal{D},t\sim\mathcal{T}}[\mathtt{sim}(g \circ f(t(\mathbf{x})), g \circ f(\mathbf{x}))]$, with 95% confidence intervals over 80k random samples in the STL10 test split [1]. Higher values mean $f(\mathbf{x})$ is more invariant to the augmentations $\mathcal{T}$.

| AugSelf (ours) | SimSiam [2] | BYOL [3] | SimCLR [4] | MoCo [5] |
|:---:|:---:|:---:|:---:|:---:|
| | $0.9263_{\pm 0.0005}$ | $0.9555_{\pm 0.0004}$ | $0.9378_{\pm 0.0006}$ | $0.9274_{\pm 0.0006}$ |
| ✓ | $0.9250_{\pm 0.0006}$ | $0.9453_{\pm 0.0004}$ | $0.9385_{\pm 0.0005}$ | $0.9280_{\pm 0.0006}$ |

## B    Hyperparameter sensitivity analysis

We simply use the same value of $\lambda$, e.g., $\lambda = 1$ for STL10 experiments, across different augmentations and different downstream tasks. One can find a better hyperparameter by tuning it on each augmentation and each downstream task, but we do not make much effort to tune it as our method is not too sensitive to hyperparameters. We here provide the sensitivity analysis to the hyperparameter $\lambda$ with varying $\lambda \in \{0.5, 1.0, 2.0\}$ under the STL10 pretraining setup. Table 2 shows that the overall transfer learning performance is not too sensitive to $\lambda$ and AugSelf clearly improves the performance in all the cases over the baseline (i.e., $\lambda = 0$).

Table 2: Linear evaluation accuracy (%) of ResNet-18 [6] pretrained by SimSiam [2] and our AugSelf with varying the hyperparameter $\lambda$.

| Pretraining objective | $\lambda$ | STL10 | CIFAR10 | CIFAR100 | Food | MIT67 | Pets | Flowers | Avg |
|---|---|---|---|---|---|---|---|---|---|
| $\mathcal{L}_{\mathtt{SimSiam}}$ | 0.0 | 85.19 | 82.35 | 54.90 | 33.99 | 39.15 | 44.90 | 59.19 | 57.09 |
| $\mathcal{L}_{\mathtt{SimSiam}} + \lambda\mathcal{L}_{\mathtt{crop}}$ | 0.5 | 85.50 | 82.81 | 55.50 | 35.19 | 42.79 | 45.94 | 61.24 | 58.42 |
| | 1.0 | 85.98 | 82.82 | 55.78 | 35.68 | 43.21 | 47.10 | 62.05 | 58.95 |
| | 2.0 | 86.36 | 82.41 | 55.29 | 36.18 | 41.91 | 47.43 | 62.28 | 58.84 |
| $\mathcal{L}_{\mathtt{SimSiam}} + \lambda\mathcal{L}_{\mathtt{color}}$ | 0.5 | 85.66 | 83.75 | 58.58 | 39.39 | 42.61 | 47.15 | 70.05 | 61.03 |
| | 1.0 | 85.55 | 82.90 | 58.11 | 40.32 | 43.56 | 47.85 | 71.08 | 61.34 |
| | 2.0 | 84.79 | 82.56 | 58.89 | 40.69 | 43.41 | 46.79 | 71.93 | 61.29 |
| $\mathcal{L}_{\mathtt{SimSiam}} + \lambda(\mathcal{L}_{\mathtt{crop}} + \mathcal{L}_{\mathtt{color}})$ | 0.5 | 86.07 | 82.67 | 57.72 | 39.92 | 43.88 | 48.86 | 70.93 | 61.43 |
| | 1.0 | 85.70 | 82.76 | 58.65 | 41.58 | 45.67 | 48.42 | 72.18 | 62.14 |
| | 2.0 | 84.56 | 83.08 | 59.49 | 41.72 | 44.50 | 49.04 | 72.35 | 62.11 |

## C Fine-tuning experiments

We fine-tune ImageNet100-pretrained models using a fine-tuning strategy, L2SP [7].[1] Following Xuhong et al. [7], we use the same set fo hyperparameters, i.e., $\beta = 0.01$, $\alpha \in \{0.001, 0.01, 0.1, 1\}$, and $lr \in \{0.005, 0.01, 0.02\}$. We evaluate the fine-tuning accuracy on five benchmarks, MIT67 [8], CUB200 [9], Food [10], Stanford Dogs [11], and Caltech256 [12]. Table 3 shows the effectiveness of our AugSelf in the fine-tuning scenarios.

Table 3: Fine-tuning accuracy (%) averaged over 5 trials with 95% confidence intervals.

| Method | MIT67 | CUB200 | Food | Standford Dogs | Caltech256 |
|---|---|---|---|---|---|
| SimSiam | $67.69_{\pm0.48}$ | $67.59_{\pm0.41}$ | $66.66_{\pm0.36}$ | $69.34_{\pm0.29}$ | $52.37_{\pm0.45}$ |
| + AugSelf (ours) | $\mathbf{69.31_{\pm0.56}}$ | $\mathbf{70.22_{\pm0.66}}$ | $\mathbf{70.19_{\pm0.18}}$ | $\mathbf{70.26_{\pm0.20}}$ | $\mathbf{54.66_{\pm0.07}}$ |

## D Robustness under perturbations

We also evaluate the robustness of the learned representations by our method. To this end, we use two types of robustness metrics: (1) adversarial robustness using the single-step fast gradient sign method (FGSM) [13] and (2) robustness to common corruptions, especially weather (fog, frost, and snow) corruptions, proposed by [14]. We here use supervised models trained on ImageNet100 for generating adversarial samples. Table 4 shows the classification accuracy on ImageNet100 under the two types of perturbations.

Table 4: Classification accuracy (%) under various perturbations.

| Method | Clean | FGSM | | | Weather corruption | | |
|---|---|---|---|---|---|---|---|
| | | $\epsilon = 1/255$ | $\epsilon = 2/255$ | $\epsilon = 4/255$ | Fog | Frost | Snow |
| SimSiam | 85.60 | 32.48 | 22.80 | 17.70 | 57.53 | 53.96 | 43.85 |
| + AugSelf (ours) | 85.40 | 32.90 | 21.60 | 16.64 | 57.42 | 53.97 | 45.65 |

We observe that our AugSelf does not significantly affect the robustness of learned representations. This result is somewhat interesting because the representations learned with AugSelf are more sensitive to diverse information than those without AugSelf. Since improving the adversarial robustness of self-supervised learning is an ongoing topic [15, 16], we believe that incorporating the idea with our framework would be an interesting research direction.

## E Datasets

Table 5 summarizes detailed descriptions of (a) pretraining datasets, (c) linear evaluation benchmarks, and (c) few-shot learning benchmarks. For linear evaluation benchmarks, we randomly choose validation samples in the training split for each dataset when the validation split is not officially provided. For few-shot benchmarks, we use the meta-test split for FC100 [17], and whole datasets for CUB200 [9] and Plant Disease [18]. The evaluation details are described in Section G.

## F Pretraining setup

### F.1 ImageNet100 pretraining

We pretrain the standard ResNet-50 [6] architecture in the ImageNet100[2] [19, 20] dataset for 500 training epochs using SimSiam [2] and MoCo [5] methods. We use a batch size of 256 and a cosine learning rate schedule without restarts [29]. Note that the pretraining setups are the same as they officially used for ImageNet pretraining described in [2, 5, 30]. In multi-GPU experiments, we use

---

[1]L2SP [7] use $\Omega(\theta) = \frac{\alpha}{2}\|\theta - \theta^0\|_2^2 + \frac{\beta}{2}\|\theta_{\mathtt{cls}}\|_2^2$ as the regularization term where $\theta^0$ is the vector of pretrained parameters (i.e., an initial point) and $\theta_{\mathtt{cls}}$ is the vector of classifier parameters.

[2]ImageNet100 is a 100-category subset of ImageNet [19]. We use the same split following Tian et al. [20].

Table 5: Dataset information.

| Category | Dataset | # of classes | Training | Validation | Test | Metric |
|---|---|---|---|---|---|---|
| (a) Pretraining | STL10 [1] | 10 | 105000 | - | - | - |
| | ImageNet100 [19, 20] | 1000 | 126689 | - | - | - |
| (b) Linear evaluation | STL10 [1] | 10 | 4500 | 500 | 8000 | Top-1 accuracy |
| | CIFAR10 [21] | 10 | 45000 | 5000 | 10000 | Top-1 accuracy |
| | CIFAR100 [21] | 100 | 45000 | 5000 | 10000 | Top-1 accuracy |
| | Food [10] | 101 | 68175 | 7575 | 25250 | Top-1 accuracy |
| | MIT67 [8] | 67 | 4690 | 670 | 1340 | Top-1 accuracy |
| | Pets [22] | 37 | 2940 | 740 | 3669 | Mean per-class accuracy |
| | Flowers [23] | 102 | 1020 | 1020 | 6149 | Mean per-class accuracy |
| | Caltech101 [24] | 101 | 2525 | 505 | 5647 | Mean Per-class accuracy |
| | Cars [25] | 196 | 6494 | 1650 | 8041 | Top-1 accuracy |
| | Aircraft [26] | 100 | 3334 | 3333 | 3333 | Mean Per-class accuracy |
| | DTD (split 1) [27] | 47 | 1880 | 1880 | 1880 | Top-1 accuracy |
| | SUN397 (split 1) [28] | 397 | 15880 | 3970 | 19850 | Top-1 accuracy |
| (c) Few-shot | FC100 [17] | 20 | - | - | 12000 | Average accuracy |
| | CUB200 [9] | 200 | - | - | 11780 | Average accuracy |
| | Plant Disease [18] | 38 | - | - | 54305 | Average accuracy |

the synchronized batch normalization following Chen and He [2]. When incorporating our AugSelf into the methods, we use $\lambda = 0.5$ and $\mathcal{A}_{\texttt{AugSelf}} = \{\texttt{crop}, \texttt{color}\}$. Note that SimSiam [2] and MoCo [5] requires 32 and 29 hours on a single RTX3090 4-GPU machine, respectively.

**SimSiam** [2]. We use a learning rate $0.05$ and a weight decay of $0.0001$. We use a 3-layer projection MLP head $g(\cdot)$ with a hidden dimension of $2048$ and an output dimension of $2048$. We use a batch normalization [31] at the last layer in the projection MLP. We use a 2-layer prediction MLP head $h(\cdot)$ with a hidden dimension of $512$ and no batch normalization at the last layer in the prediction MLP. When optimizing the prediction MLP, we use a constant learning rate schedule following Chen and He [2].

**MoCo** [5]. We use a learning rate $0.03$ and a weight decay of $0.0001$. Following an advanced version of MoCo [30, MoCo v2], we use a 2-layer projection MLP head $g(\cdot)$ with a hidden dimension of $2048$ and an output dimension of $128$. We use a batch normalization [31] at only the hidden layer. We also use a temperature scaling parameter of $0.2$, an exponential moving average parameter of $0.999$, and a queue size of $65536$.

## F.2 STL10 pretraining

We pretrain the standard ResNet-18 [6] architecture in the STL10 [1] dataset. For all methods, we use the same optimization scheme: stochastic gradient descent (SGD) with a learning rate of $0.03$, a batch size of $256$, a weight decay of $0.0005$, a momentum of $0.9$. The learning rate follows a cosine decay schedule without restarts [29]. When incorporating our AugSelf into the methods, we use $\lambda = 1.0$ and $\mathcal{A}_{\texttt{AugSelf}} = \{\texttt{crop}, \texttt{color}\}$, unless otherwise stated. We now describe method-specific hyperparameters one by one in the following.

**SimCLR** [4]. We use a 2-layer projection MLP head $g(\cdot)$ with a hidden dimension of $512$ and an output dimension of $128$. We do not use a batch normalization [31] at the last layer in the MLP. We use a temperature scaling parameter of $0.2$ in contrastive learning.

**MoCo** [5]. We use an advanced version of MoCo [30, MoCo v2] with the same projection MLP architecture as SimCLR used. Other hyperparameters are the same as the ImageNet100 setup described in Section F.1.

**BYOL** [3]. Following Grill et al. [3], we use a 2-layer projection MLP head $g(\cdot)$ with a hidden dimension of $4096$ and an output dimension of $256$. We do not use a batch normalization [31] at the last layer in the MLP. We use the same architecture for the prediction MLP head $h(\cdot)$. The exponential moving average parameter is increased starting from $0.996$ to $1.0$ with a cosine schedule following Grill et al. [3].

**SimSiam** [2]. We use a 2-layer projection MLP with a hidden dimension of $2048$ and an output dimension of $2048$. Other hyperparameters are the same as the ImageNet100 setup described in Section F.1.

**SwAV** [32]. We use a 2-layer projection MLP head $g(\cdot)$ with a hidden dimension of $2048$ and an output dimension of $128$ without batch normalization [31] at the last layer in the MLP. We use $100$ prototypes, a smoothness factor of $\epsilon = 0.05$ and a temperature scaling parameter of $\tau = 0.1$. We do not use the multi-resolution cropping technique and the additional queue storing previous batches for simplicity.

### F.3 Augmentations

In this section, we describe augmentations using PyTorch [33] notations in the following. Note that we use random cropping, flipping, color jittering, grayscale, and Gaussian blurring unless otherwise stated.

- `RandomResizedCrop`. The scale of cropping is randomly sampled from $[0.2, 1.0]$. The cropped images are resized to $96 \times 96$ and $224 \times 224$ for STL10 [1] and ImageNet [19] pretraining, respectively.
- `RandomHorizontalFlip`. This operation is randomly applied with a probability of $0.5$.
- `ColorJitter`. The maximum strengths of brightness, contrast, saturation and hue factors are $0.4, 0.4, 0.4$ and $0.1$, respectively. This operation is randomly applied with a probability of $0.8$. In Section 2, we adjust the strengths by multiplying a strength factor $s$, i.e., $s > 1$ means stronger color jittering than the default configuration while $s < 1$ means weaker color jittering.
- `RandomGrayscale`. This operation is randomly applied with a probability of $0.2$.
- `GaussianBlur`. The standard deviation is randomly sampled from $[0.1, 2.0]$. The kernel size is $9 \times 9$ and $23 \times 23$ for STL10 [1] and ImageNet [19] pretraining, respectively. This operation is randomly applied with a probability of $0.5$.
- `Rotation`. This rotates an image by $0°, 90°, 180°, 270°$ randomly. This operation is randomly applied with a probability of $0.5$ after the default geometric augmentations are applied.
- `Solarization`. This inverts each pixel value when the value is larger than a randomly sampled threshold. Formally, for an uniformly sampled threshold $\delta \sim U(0,1)$,

$$x_{\text{new}}^{(i,j)} \leftarrow \mathbb{1}[x_{\text{old}}^{(i,j)} < \delta] x_{\text{old}}^{(i,j)} + \mathbb{1}[x_{\text{old}}^{(i,j)} \geq \delta](1 - x_{\text{old}}^{(i,j)}),$$

for all pixels $(i,j)$. This operation is randomly applied with a probability of $0.5$ right after `ColorJitter` is applied.

## G Evaluation protocol

**Linear evaluation benchmarks.** We follow the same linear transfer evaluation protocol [4, 3, 34]; we train linear classifiers upon the frozen features extracted from $224 \times 224$ (or $96 \times 96$ for STL10 pretraining) center-cropped images without data augmentation. To be specific, images are first resized to $224$ pixels along the shorter side, and then cropped by $224 \times 224$ at the center of the images. Then, we minimize the $\ell_2$-regularized cross-entropy objective using L-BFGS. The regularization parameter is selected from a range of $45$ logarithmically spaced values from $10^{-6}$ to $10^{5}$ using the validation split. After selecting the best hyperparameter, we train again the linear classifier using both training and validation splits and then report the test accuracy using the model. Note that we set the maximum number of iterations in L-BFGS as $5000$ and use the previous solution as an initial point (i.e., warm start) for the next step.

**Few-shot benchmarks.** For evaluating representations in few-shot benchmarks, we simply perform logistic regression[3] using the frozen representations $f(\mathbf{x})$ and $N \times K$ support samples without fine-tuning and data augmentation in a $N$-way $K$-shot episode.

**Object localization.** For predicting bounding box information (i.e., top left coordinates and sizes of bounding boxes), we simply perform linear regression[3] using the frozen representations $f(\mathbf{x})$ and all training samples in CUB200 [9] without fine-tuning and data augmentation.

---

[3]We use the scikit-learn `LogisticRegression` and `LinearRegression` modules for logistic regression and linear regression, respectively.