# OpenReview forum: "Improving Transferability of Representations via Augmentation-Aware Self-Supervision"
_NeurIPS.cc/2021/Conference — NeurIPS 2021 Poster_

### Official Review · Reviewer_fxVi · 2021-07-16

**Rating:** 6
**Confidence:** 4

**Summary:**

This paper aims at improving the transferability of representations, especially self-supervised pre-trained representations. Motivated by the observation that augmentation-invariance may be harmful to downstream tasks, e.g., location- or color-sensitive ones, they proposed an auxiliary augmentation-aware self-supervision loss named AugSelf. AugSelf learns the difference of augmentation parameters (e.g., cropping positions, color adjustment intensities) between two randomly augmented samples. Experiments show that AugSelf helps for transfer learning tasks with two pre-trained models and a supervised learning model, as well as few-shot learning setups.


**Limitations And Societal Impact:**

Yes.

**Main Review:**

Strongness:
+ This paper is well-written and easy to follow, making it enjoyable to read.
+ The proposed method is very simple, even somewhat intuitive, by introducing an auxiliary augmentation-aware self-supervision loss.
+ Experiments on several benchmarks demonstrate the effectiveness of the proposed method.

Weakness:
- My major concern is about the claim on improving the transferability of representations. It is reasonable to achieve better performance by introducing an auxiliary augmentation-aware self-supervision loss for location- or color-sensitive tasks, but why will AugSelf work for those tasks that are not augmentation-sensitive? If not, the proposed method is not a general method to improve transferability, instead, it captures dataset-specific information.
- Another major concern is the tradeoff between preserving augmentation-aware information in learned representations with the loss of augmentation-invariant information. If the model exhausts itself to capture augmentation-aware information, will it be harmful to augmentation-invariant information? Meanwhile, how to decide the value of the hyperparameter of $\lambda$ between these two losses.
- There are two protocols to evaluate the transferability of representations: linear evaluation and fine-tuning. It seems that many fine-tuning baselines focusing on improving transferability are missing, e.g. L2SP [1], DELTA [2], BSS [3], DELTA [4].

[1] Li, X., Grandvalet, Y., and Davoine, F. Explicit inductive bias for transfer learning with convolutional networks. In ICML, 2018.
[2] Li, X., Xiong, H., Wang, H., Rao, Y., Liu, L., and Huan, J. Delta: Deep learning transfer using feature map with attention for convolutional networks. In ICLR, 2019.
[3] Chen, X., Wang, S., Fu, B., Long, M., and Wang, J. Catastrophic forgetting meets negative transfer: Batch spectral shrinkage for safe transfer learning. In NeurIPS, 2019.
[4] You, K., Kou, Z., Long, M., and Wang, J. Co-tuning for transfer learning. In NeurIPS, 2020.

- The question "when is learning invariance to a given set of augmentations harmful to representation learning?" mentioned in the Introduction section is unsolved.
- Actually, the augmentation-aware loss is defined on the same specific augmentation, instead of the whole augmentation space. Will it be possible that the proposed method be extended into different types of augmentations? If so, which kind of different information will be learned?
- Why the improvement of the proposed methods on some datasets such as STL10 in Table 9 is minor?
-Typos: Line 83 "In these methods, if h optimal, then..."




**Time Spent Reviewing:**

6

---

> ### Author Response · Authors · 2021-08-10
> **Response to Reviewer fxVi**
>
> We sincerely thank you for your helpful feedback and insightful comments. We especially appreciate several positive feedbacks of all reviewers: (1) the novel and interesting idea (Reviewer qDmv), (2) the simplicity and the wide applicability (Reviewer QC3H, wbQH, qDmv, fxVi), (3) the effectiveness across multiple benchmarks (Reviewer QC3H, wbQH, qDmv, fxVi), (4) clear write-up (Reviewer qDmv, fXvi). We address your comments and questions below.
>
> ---
> **Q1. Claim on improving the transferability of representations**
>
> **A1.** We recall that our goal is to learn general-purpose representations for both augmentation-insensitive and -sensitive downstream tasks; our method learns the representations to preserve both augmentation-invariant and augmentation-aware features. Hence, as you pointed out, if a downstream task is completely augmentation-agnostic, the effect of our axillary augmentation-aware self-supervision loss may be marginal, e.g., coarse-grained classification like STL10. However, we think a large class of downstream tasks in computer vision is location- or color-sensitive, e.g., most downstream tasks tested throughout our paper. A similar observation has been evidenced in previous self-supervised learning literature, e.g., location-sensitive [1,2] or color-sensitive [3] self-supervised objectives can learn effective features for a variety of downstream tasks in computer vision.
>
> [1] Doersch et al., Unsupervised visual representation learning by context prediction, ICCV 2015 \
> [2] Noroozi & Favaro, Unsupervised learning of visual representations by solving jigsaw puzzles, ECCV 2016 \
> [3] Zhang et al., Colorful image colorization, ECCV 2016
>
> ---
> **Q2. Tradeoff between augmentation-aware and augmentation-invariant objectives**
>
> **A2.** Thank you for mentioning the important point. First, we remark that the proposed objective function could not be harmful to learning augmentation-invariant information because we use additional separate MLP heads ($g$ and $\phi$; see Figure 1 for notations) for extracting each information from the shared representations $f(x)$. They can relax the conflict between the two objectives. To support this, we compute the cosine similarity between representations from augmented and original samples, i.e., $\mathtt{CS}=\mathbb{E}_{x\sim\mathcal{D},t\sim\mathcal{T}}[\mathtt{sim}(g\circ f(t(x)),g\circ f(x))].$ Note that this metric becomes higher as the representation $g(f(x))$ is more invariant to the augmentation $t\sim\mathcal{T}$. The following results show that AugSelf does not significantly change the cosine similarity $\mathtt{CS}$; in other words, AugSelf is not harmful to the augmentation-invariant objective.
>
> $$
> \begin{array}{ccccc}
> & \text{SimSiam} & \text{BYOL} & \text{SimCLR} & \text{MoCo} \newline \hline
> \text{w/o AugSelf} & 0.9263±0.0005 & 0.9555±0.0004 & 0.9378±0.0006 & 0.9274±0.0006 \newline
> \text{w/ AugSelf} & 0.9250±0.0006 & 0.9453±0.0004 & 0.9385±0.0005 & 0.9280±0.0006 \newline \hline
> \end{array}
> $$
>
> For the hyperparameter λ, we simply used the same value of λ, e.g., λ=1 for STL10 experiments, across different augmentations and different downstream tasks. One can find a better hyperparameter by tuning it on each augmentation and each downstream task, but we do not make much effort to tune it as our method is not too sensitive to hyperparameters. To justify this, we here provide the sensitivity analysis to the hyperparameter λ: we report the average transfer learning accuracy (%) over 7 downstream tasks (STL10, CIFAR10/100, Food, Pets, Flowers, MIT67 as we used in Section 4.2) using STL10-pretrained models with varying $\lambda\in\\{0.5,1.0,2.0\\}$. As shown below, the overall transfer learning performance is not too sensitive to λ and AugSelf clearly improves the performance in all the cases over the baseline (λ=0).
>
> $$
> \begin{array}{lcccc}
> \text{Objective} & \lambda=0 & \lambda=0.5 & \lambda=1.0 & \lambda=2.0 \newline \hline
> \mathcal{L}_\mathtt{SimSiam}+\lambda\mathcal{L}_\mathtt{crop} &
> 57.09 & 58.42 & 58.95 & 58.84 \newline
> \mathcal{L}_\mathtt{SimSiam}+\lambda\mathcal{L}_\mathtt{color} &
> 57.09 & 61.03 & 61.34 & 61.29 \newline
> \mathcal{L}_\mathtt{SimSiam}+\lambda(\mathcal{L}_\mathtt{crop}+\mathcal{L}_\mathtt{color}) &
> 57.09 & 61.43 & 62.14 & 62.21 \newline \hline
> \end{array}
> $$
>
> We will incorporate these additional experiments and discussions in the final version.
>
> ---
> **Q3. Fine-tuning experiments**
>
> **A3.** Thank you for the important suggestion. Following your suggestion, we fine-tune ImageNet100-pretrained models using L2SP [1] on five datasets, MIT67, CUB200, Food, Stanford Dogs, and Caltech256. We use the same hyperparameter setup as [1].
>
> $$
> \begin{array}{lccccc}
> & \text{MIT67} & \text{CUB200} & \text{Food} & \text{Stanford Dogs} & \text{Caltech256} \newline \hline
> \text{SimSiam} & 67.69±0.48 & 67.59±0.41 & 66.66±0.36 & 69.34±0.29 & 52.37±0.45 \newline
> \text{+ AugSelf} & 69.31±0.56 & 70.22±0.66 & 70.19±0.18 & 70.26±0.20 & 54.66±0.07 \newline \hline
> \end{array}
> $$
>
> As reported above, our AugSelf is effective in not only linear evaluation, but also fine-tuning. Since AugSelf is orthogonal to fine-tuning, we believe that AugSelf can be incorporated with other fine-tuning methods. We will conduct more fine-tuning experiments and incorporate the results in the final draft.
>
> [1] Li et al., Explicit Inductive Bias for Transfer Learning with Convolutional Networks, ICML 2018
>
> ---
> **Q4. “when is learning invariance to a given set of augmentations harmful to representation learning?” is unsolved.**
>
> **A4.** We think L39-41 (and also Figure 2 and L89-L100) answered the question: learning augmentation-invariant representations might hurt its performance when transferring the learned representations to some augmentation-sensitive downstream tasks, in other words, it could be harmful to representation learning. We will further clarify the paragraph in the final version.
>
> ---
> **Q5. Is it possible that AugSelf be extended into different types of augmentations?**
>
> **A5.** Yes, AugSelf can be extended to various augmentation types (and also various combinations) as discussed around Table 6-7 and the “different augmentations” paragraph in Section 4.2. The learned information will rely on the augmentation types, for example, AugSelf with rotations (i.e., predicting rotation degrees) could learn semantic information (e.g., location, pose, etc) of objects as shown in [1].
>
> [1] Gidaris et al., Unsupervised Representation Learning by Predicting Image Rotations, ICLR 2018
>
> ---
> **Q6. Why is the improvement on some downstream tasks minor?**
>
> **A6.** As we responded in Q1, the performance gain of AugSelf may depend on characteristics, e.g., augmentation sensitivity, of each downstream task. For example, STL10 has 10 categories (airplane, bird, car, cat, deer, dog, horse, monkey, ship, truck) and one can easily discriminate them using only part of an image without color information. Although our AugSelf is more effective in more augmentation-sensitive tasks, we would like to emphasize that it is, at least, not harmful to augmentation-invariant tasks; hence, one can use AugSelf without prior knowledge on downstream tasks.
>
> ---
> **Q7. Editorial comments**
>
> **A7.** Thank you for the suggestion to improve our writing. We will incorporate the suggested editorial comments in the final draft.

---

> > ### Comment · Reviewer_fxVi · 2021-08-18
> > **Elegant Response**
> >
> > Thanks for providing such an elegant response, especially demonstrating that AugSelf will not be harmful to augmentation-invariant tasks. This observation makes it easy to tailor AutoSelf into many downstream tasks without any prior knowledge. I appreciate the authors for addressing my major concerns and will raise my score to 6 (Marginally above the acceptance threshold). Hope that the feedback provided in the rebuttal will be carefully incorporated into the main body of the paper.

---

> > > ### Author Response · Authors · 2021-08-27
> > > **Thank you for the response**
> > >
> > > We are happy to hear that our rebuttal addressed your concerns well.
> > >
> > > Thank you again for the valuable suggestions and comments to add, which we will incorporate in the revision to strengthen our paper.
> > >
> > > If you have any remaining suggestions or concerns, please let us know!
> > >
> > > Best, Authors.

---

### Official Review · Reviewer_qDmw · 2021-07-16

**Rating:** 7
**Confidence:** 4

**Summary:**

This paper presents AugSelf, an auxiliary self-supervised task in contrastive learning to improves the transferability of learned representations. The task predicts the difference of augmentation parameters between two randomly augmented samples, and therefore forces the representation to preserve the information which could be potentially discarded by augmentations. Extensive experiments demonstrate the effectiveness of the proposed method.


**Limitations And Societal Impact:**

Yes.

**Main Review:**

Originality: The observation that “learning representations with an augmentation-invariant objective might hurt its performance in downstream tasks” and the idea of preserving the information that is removed by the augmentations to improve transferability is already presented in [1]. However, the idea of adding an auxiliary task to preserve the information is novel and interesting.

Quality:
The proposed method of predicting the augmentations to preserve information is novel and interesting. The method is simple and can be easily adapted to different contrastive learning framework and is applicable to a wide range of common quantifiable augmentations. The experiments and ablation studies are also extensive.

Minor points:
1.	the observation that “learning representations with an augmentation-invariant objective might hurt its performance in downstream tasks” should not be claimed as part of the contribution since [1] already observed it.
2.	The checklist is not well formatted. The instructions are not deleted and there are still placeholders. Also, the error bars and standard deviation is missing, which could be >0.5 as shown in [1].

Clarity: The paper is well written and easy to follow.

Significance: the paper proposes a simple method to preserve information that could be potentially removed by the strong augmentations of contrastive learning. Although the idea of preserving the information removed by augmentations to improve transferability is not new, the method is simple and effective to preserve the information and its performance on multiple datasets are consistently good.

[1] Xiao, Tete, et al. "What should not be contrastive in contrastive learning." arXiv preprint arXiv:2008.05659 (2020).


**Time Spent Reviewing:**

2 hours

---

> ### Author Response · Authors · 2021-08-10
> **Response to Reviewer qDmw**
>
> We sincerely thank you for your helpful feedback and insightful comments. We especially appreciate several positive feedbacks of all reviewers: (1) the novel and interesting idea (Reviewer qDmv), (2) the simplicity and the wide applicability (Reviewer QC3H, wbQH, qDmv, fxVi), (3) the effectiveness across multiple benchmarks (Reviewer QC3H, wbQH, qDmv, fxVi), (4) clear write-up (Reviewer qDmv, fXvi). We address your comments and questions below.
>
> ---
> **Q1. Comments on the observation that “learning representations with an augmentation-invariant objective might hurt its performance in downstream tasks”**
>
> **A1.** Thank you very much for the detailed comment. We agree that the observation was also claimed in [1], but we would like to note that we additionally conducted supporting experiments, for example, observations on mutual information between representations and color information as shown in Figure 2(a). We will clarify the contribution paragraph in the Introduction section in the final draft.
>
> [1] Xiao et al., What Should Not Be Contrastive in Contrastive Learning, ICLR 2021
>
> ---
> **Q2. Formatting and standard deviation**
>
> **A2.**
> Thank you very much for the comment. We will revise the checklist in the final draft. We also will add the standard deviations into the final draft. We would like to note that the standard deviation is relatively smaller than our gains as shown in Table 2 and 3.

---

### Official Review · Reviewer_wbQH · 2021-07-17

**Rating:** 6
**Confidence:** 4

**Summary:**

This paper proposes a self-supervision method with learning the difference of augmentation parameters as the pretext task to improve the transferability of learned representations.

**Limitations And Societal Impact:**

I don't see any potential negative societal impact of their work.

**Main Review:**

Strength:
+ The paper proposes a new self-supervision method to improve the transferability of learned representations. The method is simple and clear.
+ The proposed method consistently improves the unsupervised learning methods.

Weakness:
- There are no other sota representation learning methods, especially other self-supervision related methods with different design of pretext tasks for comparison.


Question:
Does the augmentation help with the robustness of the learned representation under perturbations?

**Time Spent Reviewing:**

2 hours.

---

> ### Author Response · Authors · 2021-08-10
> **Response to Reviewer wbQH**
>
> We sincerely thank you for your helpful feedback and insightful comments. We especially appreciate several positive feedbacks of all reviewers: (1) the novel and interesting idea (Reviewer qDmv), (2) the simplicity and the wide applicability (Reviewer QC3H, wbQH, qDmv, fxVi), (3) the effectiveness across multiple benchmarks (Reviewer QC3H, wbQH, qDmv, fxVi), (4) clear write-up (Reviewer qDmv, fXvi). We address your comments and questions below.
>
> ---
> **Q1. There are no other SOTA methods.**
>
> **A1.** We believe that we already demonstrated the wide compatibility of our method with various SOTA methods, SimSiam, MoCo, SimCLR, BYOL, and SwAV, as shown in Table 1 and Table 9. More importantly, our method is an auxiliary objective and orthogonal to other self-supervised learning objectives, so it can be easily incorporated with the objectives. For example, Barlow Twins [1] and DINO [2], published concurrently to our submission, use two augmented samples for each instance; hence, one can add our auxiliary objective into the methods as we did in our paper. We will add the respective discussion to the final draft.
>
> [1] Zbontar et al., Barlow Twins: Self-Supervised Learning via Redundancy Reduction, 2021 \
> [2] Caron et al., Emerging Properties in Self-Supervised Vision Transformers, 2021
>
> ---
> **Q2. Robustness under perturbations.**
>
> **A2.** Thank you for your interesting suggestion. To evaluate the robustness of the learned representations, we evaluate two types of robustness metrics: (1) adversarial robustness using the single-step fast gradient sign method (FGSM) and (2) robustness to common corruptions, especially weather (fog, frost, snow) corruptions, proposed by [1]. We here use supervised models trained on ImageNet100 for generating adversarial samples.
>
> $$
> \begin{array}{lccccccc}
> & \text{Clean} & \text{FGSM} (\varepsilon=\frac{1}{255}) & \text{FGSM} (\varepsilon=\frac{2}{255}) & \text{FGSM} (\varepsilon=\frac{4}{255}) & \text{Fog} & \text{Frost} & \text{Snow} \newline \hline
> \text{Supervised} & 85.60 & 32.48 & 22.80 & 17.70 & 57.53 & 53.96 & 43.85 \newline
> \text{+ AugSelf} & 85.40 & 32.90 & 21.60 & 16.64 & 57.42 & 53.97 & 45.65 \newline \hline
> \end{array}
> $$
>
> We observe that our AugSelf does not significantly affect the robustness of learned representations. This result is somewhat interesting because the representations learned with AugSelf are more sensitive to diverse information than those without AugSelf. Since improving the adversarial robustness of self-supervised learning is an ongoing topic [2,3], we believe that incorporating the idea with our framework would be an interesting research direction.
>
> [1] Hendrycks & Dietterich, Benchmarking Neural Network Robustness to Common Corruptions and Perturbations, ICLR 2019 \
> [2] Kim et al., Adversarial Self-Supervised Contrastive Learning, NeurIPS 2020 \
> [3] Ho & Vasconcelos, Contrastive Learning with Adversarial Examples, NeurIPS 2020

---

### Official Review · Reviewer_QC3H · 2021-07-20

**Rating:** 6
**Confidence:** 4

**Summary:**

This paper proposes a simple regularization for self-supervised/supervised representation learning, which aims to benefit downstream tasks. Main idea is to make the network to be variant to different augmentation. Extensive experimental results are reported to show the effectiveness.

**Limitations And Societal Impact:**

Yes.

**Main Review:**

1. The idea is simple and seems to provide good performance gains. However, I am curious what's the main learning objective of the main paper. The main loss tries to learn augmentation-invariant representation while the AugSelf tries to learn augmentation-variant representation. These two losses are contradicting. What do we want the network to learn? To be invariant or variant to augmentation. If some augmentations are bad for downstream tasks, such as color jittering and Gaussian blurring, can we just remove these negative augmentation in the main loss? Maybe just removing these augmentation is sufficient to bring large performance gains. Adding the regularization will make things complex. So I suggest to add ablation studies on this: main loss without color/blurring, and more importantly provide more insights about why your regularized method could work.

2. How can you determine λ for each augmentation? Can you provide the sensitivity analysis to these hyperparamters.

**Time Spent Reviewing:**

4 hours

---

> ### Author Response · Authors · 2021-08-10
> **Response to Reviewer QC3H**
>
> We sincerely thank you for your helpful feedback and insightful comments. We especially appreciate several positive feedbacks of all reviewers: (1) the novel and interesting idea (Reviewer qDmv), (2) the simplicity and the wide applicability (Reviewer QC3H, wbQH, qDmv, fxVi), (3) the effectiveness across multiple benchmarks (Reviewer QC3H, wbQH, qDmv, fxVi), (4) clear write-up (Reviewer qDmv, fXvi). We address your comments and questions below.
>
> ---
>
> **Q1. Contradiction between augmentation-invariant and augmentation-variant objectives**
>
> **A1.** Thank you for mentioning the important point and suggesting interesting ablation studies. Although the two objectives have different goals, they are not necessarily contracting; indeed, they encourage the representations $f(x)$ to preserve both augmentation-invariant and augmentation-variant features. This is doable because we use separate MLP heads, $g$ and $\phi$, for the two objectives. The MLP heads are learned to extract augmentation-invariant and augmentation-aware information from the shared representations $f(x)$, respectively. Such an architectural approach (e.g., multi-branch) for multiple objectives has been widely used in multi-task learning literature, e.g., [1-3].
>
> As you suggested, one can attempt to eliminate an augmentation-invariant property from representations by removing the augmentation in the pretraining stage. However, this strategy could significantly affect the quality of representations learned by augmentation-invariant objectives (e.g., see Table 17 in [3]), which leads to a performance drop even in augmentation-sensitive downstream tasks. To verify this, we pretrain SimSiam without the ColorJittering augmentation on STL10 and evaluate the learned representations in STL10, CIFAR10/100, Food, MIT67, Pets, and Flowers benchmarks as we did in Section 4.2. As shown below, removing ColorJittering degrades performance significantly, even in color-sensitive tasks, Food and Flowers. In contrast, our approach can enjoy the advantages of the recent progress in augmentation-invariant objectives such as SimSiam (see the last row) since AugSelf does not modify the objectives.
>
> $$
> \begin{array}{lccccccc}
> & \text{STL10} & \text{CIFAR10} & \text{CIFAR100} & \text{Food} & \text{MIT67} & \text{Pets} & \text{Flowers} \newline
> \hline
> \text{SimSiam w/o ColorJittering Aug.} & 78.53 & 76.58 & 45.45 & 29.09 & 30.77 & 34.17 & 46.38 & \newline
> \text{SimSiam}                  & 85.19 & 82.35 & 54.90 & 33.99 & 39.15 & 44.90 & 59.19 \newline
> \text{SimSiam + AugSelf}  & 85.70 & 82.76 & 58.65 & 41.58 & 45.67 & 48.42 & 72.18 \newline \hline
> \end{array}
> $$
>
> As another support, we also compute the cosine similarity between representations from augmented and original samples, i.e.,
> $\mathtt{CS}=\mathbb{E}_{x\sim\mathcal{D},t\sim\mathcal{T}}[\mathtt{sim}(g\circ f(t(x)),g\circ f(x))].$ Note that this metric becomes higher as the representation $g(f(x))$ is more invariant to the augmentation $t\sim\mathcal{T}$. The following results show that AugSelf does not significantly change the cosine similarity $\mathtt{CS}$; in other words, AugSelf is not harmful to the augmentation-invariant objective.
>
> $$
> \begin{array}{ccccc}
> & \text{SimSiam} & \text{BYOL} & \text{SimCLR} & \text{MoCo} \newline \hline
> \text{w/o AugSelf} & 0.9263±0.0005 & 0.9555±0.0004 & 0.9378±0.0006 & 0.9274±0.0006 \newline
> \text{w/ AugSelf} & 0.9250±0.0006 & 0.9453±0.0004 & 0.9385±0.0005 & 0.9280±0.0006 \newline \hline
> \end{array}
> $$
>
> We will incorporate these additional experiments and discussions in the final version.
>
> [1] Long et al., Learning Multiple Tasks with Multilinear Relationship Networks, NIPS 2017 \
> [2] Misra et al., Cross-stitch Networks for Multi-task Learning, CVPR 2016 \
> [3] Xiao et al., What Should Not Be Contrastive in Contrastive Learning, ICLR 2021 \
> [4] Grill et al., Bootstrap Your Own Latent: A New Approach to Self-supervised Learning, 2020
>
> ---
> **Q2. Sensitivity analysis to the hyperparameter λ**
>
> **A2.** We simply used the same value of λ, e.g., λ=1 for STL10 experiments, across different augmentations and different downstream tasks. One can find a better hyperparameter by tuning it on each augmentation and each downstream task, but we do not make much effort to tune it as our method is not too sensitive to hyperparameters, as shown in what follows.
>
> As you suggested, we here provide sensitivity analysis to the hyperparameter λ: we report the average transfer learning accuracy (%) over 7 downstream tasks (STL10, CIFAR10/100, Food, Pets, Flowers, MIT67 as we used in Section 4.2) using STL10-pretrained models with varying $\lambda\in\\{0.5,1.0,2.0\\}$. As shown below, the overall transfer learning performance is not too sensitive to λ and AugSelf clearly improves the performance in all the cases over the baseline (λ=0). We will add these results into the final draft.
>
> $$
> \begin{array}{lcccc}
> \text{Objective} & \lambda=0 & \lambda=0.5 & \lambda=1.0 & \lambda=2.0 \newline \hline
> \mathcal{L}_\mathtt{SimSiam}+\lambda\mathcal{L}_\mathtt{crop} &
> 57.09 & 58.42 & 58.95 & 58.84 \newline
> \mathcal{L}_\mathtt{SimSiam}+\lambda\mathcal{L}_\mathtt{color} &
> 57.09 & 61.03 & 61.34 & 61.29 \newline
> \mathcal{L}_\mathtt{SimSiam}+\lambda(\mathcal{L}_\mathtt{crop}+\mathcal{L}_\mathtt{color}) &
> 57.09 & 61.43 & 62.14 & 62.21 \newline \hline
> \end{array}
> $$

---

> > ### Comment · Reviewer_QC3H · 2021-08-25
> > **Further feedbacks**
> >
> > Thanks for the detailed explanation. But I am still confused about how the multi-heads, g and \phi can effect the feature extractor f, which is used for transfer learning. I know the g can learn augmentation invariant features while \phi learn augmentation variant features. But do you want f to be more augmentation invariant or not? If you want it to be less augmentation invariant, may be it's just necessary to reduce the augmentation ratio for certain augmentation approaches?

---

> > > ### Author Response · Authors · 2021-08-27
> > > **Thank you for further feedbacks**
> > >
> > > Many thanks for providing your additional feedback.
> > >
> > > We first emphasize that we want $f(x)$ to learn (or contain) both augmentation-invariant and augmentation-aware information (or features) in the input $x$. To this end, we train $g$ and $\phi$ to extract each information from $f(x)$, respectively, i.e., we want the functions $g(f(t_1(x)))$ and $\phi(f(t_1(x)),f(t_2(x)))$ to be invariant and variant with respect to augmentation $t_1$ (and $t_2$), respectively. Here, if the shared network $f$ has a limited capacity (e.g., few parameters or dimension), the two training objectives (for $g$ and $\phi$) may interfere with each other, i.e., $f(x)$ might become less invariant (or contain less augmentation-invariant information), as you pointed out. However, our choice $f$ of deep neural network (DNN) in our experiments does not suffer from the issue (i.e., DNN is highly expressive), so our goal is achievable with a negligible loss of augmentation-invariant information, as empirically shown in the previous response (see the second experiment in A1).
> > >
> > > Also, as you suggested, one may learn representations to be less invariant to a specific augmentation by reducing its augmentation strength. However, this strategy is not an effective way for learning augmentation-aware information, as evidenced in our paper (see Figure 2). As shown in Figure 2(b) and 2(c), the gains from the strategy are marginal in augmentation-sensitive downstream tasks. In contrast, our auxiliary tasks can encourage $f(x)$ to learn the augmentation-aware information more directly; this leads to significant gains in the downstream tasks.
> > >
> > > We hope this clarifies your concerns. If that is not the case, please let us know again!

---

### Author Response · Authors · 2021-08-18
**A gentle reminder**

Dear Reviewers,

Thank you for your time and efforts in reviewing our paper.

We kindly remind that we are more than one week into the discussion period. We believe that we sincerely and successfully address your concerns/questions/misunderstandings/suggestions, with the results of the supporting experiments.

If you have any further concerns or questions, please do not hesitate to let us know.

Thank you very much! Authors

---

### Decision · Program_Chairs · 2021-09-27

**Decision:**

Accept (Poster)

**Comment:**

This paper aims to improve the transferability of representations via augmentation-aware self-supervision, by preserving both augmentation-invariant and augmentation-aware information. The main finding is that augmentation-variant information may be relevant to downstream tasks, which is interesting and inspiring for the self-supervised learning community. The rebuttal is informative and relevant, and after the discussion all reviewers unanimously recommended acceptance. Reviewers also pointed out some important directions for improvement, such as a larger set of augmentations for pre-training, more sophisticated fine-tuning methods, and a clear elaboration on the relationship between augmentation invariance and awareness. Authors shall guarantee to include the rebuttal material into their future draft.